# Investigation into Performance of Multilayer Composite Nano-Structured Cr-CrN-(Cr$_{0.35}$Ti$_{0.40}$Al$_{0.25}$)N Coating for Metal Cutting Tools

**Sergey Grigoriev [1], Alexey Vereschaka [2,*] , Alexander Metel [1] , Nikolay Sitnikov [3] , Filipp Milovich [4], Nikolay Andreev [4], Svetlana Shevchenko [5] and Yulia Rozhkova [5]**

1 Department of High-Efficiency Processing Technologies, Moscow State Technological University STANKIN, 127055 Moscow, Russia; s.grigoriev@stankin.ru (S.G.); a.metel@stankin.ru (A.M.)
2 Department of Mechanical Engineering, Moscow State Technological University STANKIN, 127055 Moscow, Russia
3 Department of Solid State Physics and Nanosystems, National Research Nuclear University MEPhI, 115409 Moscow, Russia; sitnikov_nikolay@mail.ru
4 Materials Science and Metallurgy Shared Use Research and Development Center, National University of Science and Technology MISIS, 119049 Moscow, Russia; filippmilovich@mail.ru (F.M.); andreevn.misa@gmail.com (N.A.)
5 Material Properties Research Laboratory, Moscow State Technological University STANKIN, 127055 Moscow, Russia; s.shevchenko@lism-stankin.ru (S.S.); u.rozhkova@lism-stankin.ru (Y.R.)
* Correspondence: ecotech@rambler.ru; Tel.: +7-916-910-0413

**Abstract:** This paper deals with the Cr-CrN-(Cr$_{0.35}$Ti$_{0.40}$Al$_{0.25}$)N coating. It has a three-layered architecture with a nano-structured wear-resistant layer. The studies involved the investigation into the microstructure (with the use of SEM and TEM), elemental and phase composition (XRD and SAED patterns), wear process pattern in scratch testing, crystal structure, as well as the microhardness of the coating. Cutting tests of tools with the above coating were carried out in dry turning of steel 1045 at cutting speeds of $v_c$ = 200, 250, and 300 m·min$^{-1}$. The comparison included uncoated tools and tools with the commercial TiN and (Ti,Al)N coatings with the same thickness. The tool with the Cr-CrN-(Cr$_{0.35}$Ti$_{0.40}$Al$_{0.25}$)N coating showed the longest tool life at all the cutting speeds under consideration. Meanwhile, a tool with the coating under study can be recommended for use in turning constructional steel at the cutting speed of $v_c$ = 250 m·min$^{-1}$. At this cutting speed, a tool shows the combination of a rather long tool life and balanced wear process, without any threat of catastrophic wear.

**Keywords:** filtered cathodic vacuum arc deposition (FCVAD) coatings; tool life; thermostability; nanostructured coatings

## 1. Introduction

The (Ti,Cr,Al)N coatings with different ratios of the composition elements were investigated in a large number of studies [1–13], which showed the high efficiency of the coatings under study due to their good resistance to oxidation and wear at high temperatures. This combination of properties opens up wide opportunities for their use as a wear-resistant coating for metal cutting tools, including those operated at high cutting speeds at high temperatures.

An increase in temperature initiates macro- and micro-fractures on contact pads of the tool. With an increase in the cutting speed, there is a tendency for the isotherms of maximum temperatures

to move towards the cutting edge and to the flank face of the tool [1–4]. This is determined by a change in the contact characteristics: A decrease in the total contact length on the rake face and a slight increase in the length of total contact on the flank face, an increase in the shear angle and a decrease in secondary deformations. The pattern of the change in an isotherm in the cutting part of the tool also explains a decrease in intensity of the crater formation for coated tools. Thus, in terms of estimating the thermal state of the cutting tool, a greater effect from the coating can be achieved at high cutting speeds. When assessing the effect of the coating on the thermal state of the cutting zone and the cutting tool, it is necessary to take into account the very complex changes introduced by the coating to the contact processes, the chip formation mechanism and the formation of contact areas, the buildup process, and the change in plastic shear resistance. Thus, a favourable redistribution of heat flows in cutting with the coated tool and is an extremely important control function of the coating in terms of improving cutting performance and increasing tool efficiency [1,4].

In particular, Yamamoto et al. [5] found that the coating under study showed good oxidation resistance till 1000 °C, i.e., 200 °C higher than the (Ti,Al)N coating. It was also found [6] that at a temperature of about 900 °C, a cubic structure in the coating under study transformed into a hexagonal one, and that led to a decrease in the coating hardness by about 4 GPa. It was found [7] that the (Ti,Cr,Al)N nanolayer had a cubic structure, while the (Al,Si)N nanolayer had a hexagonal structure. The hardness of the coating and its thermostability increased when the value thickness of the binary nanolayer $\lambda$ decreased. The cutting tests conducted in end milling of H13 steel (HRC 50) showed [8] a significant increase (by two times) in tool life compared to the traditional (Ti,Al)N coating. During the consideration of a tool with the specified coating for machining workpieces of H13 (HRC 50–52) tool steel, as well as TiAl6V4 alloy and Ni-based superalloy, it was found [9] that the (Ti,Cr,Al)N coating showed the best result in the machining of hardened steel, with high temperatures in the cutting zone. Meanwhile, the (Ti,Al)N coating showed the best result in machining of TiAl6V4 alloy and Ni-based superalloy. The authors explain [9] the difference by cutting conditions: The value of resistance to oxidation and hardness at high temperatures in the machining of hardened steel and the value of plasticity and resistance to brittle fracture in machining of the above alloys. The study [10] revealed the presence of the $TiCrN_2$ phase in the composition of the specified coating, as well as the solid solution of Cr and Al in titanium nitride of (Ti,Cr,Al)N, with the lattice parameter of $a$ = 4.244–4.223 Å (depending on the Cr content). It was found [11,12] that at room temperature, the (Ti,Al,Cr)N and (Ti,Al)N coatings showed similar tribological properties, but at a temperature of 650 °C, the Cr-containing coatings showed the best properties due to the formation of chromium oxide tribofilm, superior to similar films based on Al and Ti oxides. Meanwhile, the tribological properties increased with an increase of Cr content in the coating. The cutting tests showed [11,12] that at low cutting speeds, a coating without Cr content demonstrated the best result due to its higher hardness and, respectively, wear resistance. However, with an increase in cutting speed, the tools with Cr-containing coatings showed a noticeable decrease in cutting force and a longer tool life due to an active formation of oxide tribofilms. The study [13] revealed the formation of the oxides of $Cr_2O_3$, $\alpha$-$Al_2O_3$, $SiO_2$ and rutile-$TiO_2$ at temperatures of 600–1000 °C. The oxidation tests carried out at temperatures of 500–900 °C showed [14] the best oxidation resistance of the (Cr,Ti,Al)N coating in combination with good structural stability at high temperatures. It was found that the high oxidation resistance was provided due to the formation on surface of the $Cr_2O_3$ and $Al_2O_3$ films, effectively preventing further diffusion of oxygen into the coating. At a temperature of 900 °C, the coating under study retained its crystal structure, surface morphology, hardness, and wear resistance. It was found [15] that while at room temperature, the maximum hardness was observed for the coatings with high Cr content (38–39 at.%), then at a temperature of 900 °C, the maximum hardness was demonstrated by the (Ti,Al)N coating, and at a temperature of 1100 °C—by the coatings with low Cr content (7 and 28 at.%). Meanwhile, the coatings with high Ti content showed the lowest resistance to oxidation at high temperatures due to the lowest protective properties of the $TiO_2$ oxide film compared to $Cr_2O_3$ and $Al_2O_3$. It was found [16] that the coating structure (fcc-(Cr,Al)N and h-(Al,Si)N phases)

did not fail until reaching a temperature of 900 °C, and at higher temperatures, the fcc-CrN phase decomposed with the formation of the h-$Cr_2N$, cubic Cr, and $Al_{80}Cr_{20}$ phases. The coating retained its high resistance to oxidation until the temperature of 900 °C. Introduction of Cr in the coating content inhibited the transformation of an-$TiO_2$ into r-$TiO_2$, and also inhibited the growth of a porous sublayer, rich in titanium oxides, and thus increased the oxidation resistance of the coating [14]. The best results at room temperature and at a temperature of 400 °C were demonstrated by the $(Cr_{0.40}Ti_{0.22}Al_{0.38})N$ coating [17,18]. Lin et al. [19] found that the $(Cr_{0.61}Ti_{0.10}Al_{0.29})N$ coating showed the maximum hardness (40 GPa) and wear resistance, as well as the minimum friction coefficient both at room temperature and at temperatures of 600–800 °C. The coating also showed the resistance to oxidation at temperatures of up to 1000 °C. With an increase in the Ti content in the coating, the specified properties decreased. Lu et al. [20] considered the Cr-CrN-(Cr,Ti,Al)N coating. The outer wear-resistant layer with a thickness of about 2 μm had a columnar structure. The cutting tests conducted in end milling of hardened steel P20 (HRC 45) showed a significant increase in tool life compared to an uncoated tool.

Thus, summarizing the research experience available, it is possible to draw the following conclusions:

- The (Cr,Ti,Al)N coating can be effectively used to improve tool life of cutting tools under conditions of high temperatures in the cutting zone and active abrasive wear.
- An increase in Ti (over 25–28 at.%) in the coating composition leads to an increase in its hardness and wear resistance, especially at high temperatures, but worsens its oxidation resistance.
- Accordingly, with an increase in the Cr content, the friction coefficient decreases and the resistance to oxidation at high temperatures increases.
- The use of nanolayer coatings makes it possible to further improve the performance properties of cutting tools.

## 2. Materials and Methods

The multilayer Cr-CrN-$(Cr_{0.35}Ti_{0.40}Al_{0.25})N$ coating with a nano-structured outer wear-resistant layer was taken as an object of this study. It was anticipated that such a coating would provide a balanced combination of wear resistance, thermostability, resistance to high temperature oxidation, and brittle fracture. The three-layer architecture of the coating was used (adhesive layer–transition layer–wear-resistant layer), and considered in previous works [21–25].

For coating deposition, the filtered cathodic vacuum-arc deposition (FCVAD) VIT-2 unit (MSTU STANKIN-IKTI RAN, Moscow, Russia) [26–31] was used. These coatings were deposited on carbide inserts.

The deposition process parameters ($n$—rotation speed of a turntable, $I_{Al}$—Al cathode arc current, $I_{Ti}$—Ti cathode arc current, $I_{Cr}$—Cr cathode arc current, $U_b$—value of the displacement potential on the substrate and $P_N$—the pressure of the reaction gas (nitrogen)) are presented in Table 1. During the deposition process, three-cathode systems with Cr (99.97%), Ti (99.98%) and Al (98.2% + 1% Si) cathodes were used.

**Table 1.** Deposition parameters for the coatings under study.

| Architecture of Coating | Parameters for Deposition Process | | | | | |
|---|---|---|---|---|---|---|
| | $n$ (rev·min$^{-1}$) | $I_{Al}$ (A) | $I_{Ti}$ (A) | $I_{Cr}$ (A) | $U_b$ (V) | $P_N$ (Pa) |
| Cr-CrN-$(Cr_{0.35}Ti_{0.40}Al_{0.25})N$ | 1.2 | 170 | 60 | 75 | −160 | 0.4 |

The cutting tests were carried out in dry cutting of a workpiece made of steel 1045 to study the effect of cutting speed on tool life of a tool with the coating under study, at three different cutting speeds: $v_c$ = 200, 250, and 300 m·min$^{-1}$ ($f$ = 0.2 mm·rev$^{-1}$; $a_p$ = 1.0 mm). Conditions of the cutting tests: Turning material: Steel AISI 1045 (HB 180), workpiece sizes: diameter 250 mm, length 400 mm; cutting tool: Carbide inserts SNUN ISO 1832:2017 [32]. (WC + 12% TiC + 6% Co, Kirovgrad Carbide Plant, Russia); tool geometry: The rake angle $\Upsilon$ = −7°; the clearance angle $\alpha$ = 7°; the tool cutting edge inclination $\lambda$ = 0; the nose radius $R$ = 0.8 mm; The flank wear $VB_{max}$ = 0.4 mm was assumed as the

failure criterion. Based on the data available [33–35], it could be predicted that at lower cutting speeds, the abrasive wear was essential and that predetermined the need to use a coating with the maximum hardness and wear resistance. As cutting speed increases, a temperature in the cutting zone rises and adhesive, oxidative, and diffusion components of the wear mechanism come to the fore. Accordingly, the coating used in such conditions should have good thermo- and oxidation resistance in combination with the resistance to brittle fracture, reduced tendency to adhesion to the material being machined, as well as good barrier properties against diffusion.

An uncoated tool and tools with the TiN and (Ti,Al)N coatings with similar thickness of about 3 μm were used as an object of comparison.

For microstructural studies of samples, a SEM FEI Quanta 600 FEG (Materials & Structural Analysis Division, Hillsboro, OR, USA) was used. The microhardness (HV) of coatings was measured by using the method of Oliver and Pharr [36], at a fixed load of 20 mN. The adhesion characteristics were studied on a Nanovea scratch tester (Micro Scratch, Nanovea, Irvine, CA, USA). The tests were carried out with the load linearly increasing from 0.05 to 40 N [37].

To determine the phase composition of the coating, X-ray structural phase analysis was used. Diffraction patterns were taken on a PANalytical Empyrean Series 2 X-ray diffractometer (Malvern Panalytical Ltd., Malvern, UK) using monochromatic CuKα radiation, in an asymmetric geometry (sliding beam) using a parallel beam. The analysis of the phase composition was carried out using the software PANalytical High Score Plus and the database ICCD PDF-2.

The structure of the crystals was studied under a JEM 2100 (JEOL, Tokyo, Japan) microscope at an acceleration voltage of 200 keV. Samples for TEM study were prepared by Strata FIB 205 System (FEI Company, Hillsboro, OR, USA). Element analyses were carried out by EDX INCA Energy (Oxford Instruments, Oxfordshire, UK).

## 3. Results

Following the study of the coating microstructure (Figures 1 and 2), it was possible to make the following conclusions:

- The total coating thickness was about 3 μm, the thickness of the transition CrN layer was about 300 nm (Figure 1), the thickness of the adhesive Cr layer was about 20 nm (Figure 2, Area I).
- The wear-resistant layer included 24 binary nanolayers. The thickness of subnanolayers in the wear-resistant layer was 1.5–5.0 nm (Figure 2, Area III), with the thickness of the binary nanolayer of $\lambda = 102$ nm.
- There was an area with low Al content between the transition and wear-resistant layers. The formation of the above area was connected with the provision of a gradient transition from the coating containing no Al to the coating with the Al content of about 25 at.% (Figures 1c and 2, Area II).
- The following commercial coatings were selected as objects of comparison in the study of cutting properties: TiN monolayer coating (thickness was about 3 μm, Figure 1a) and multilayer nanostructured coating TiN-(Ti$_{75}$Al$_{25}$)N with a total thickness of about 3 μm (Figure 1b). These coatings are widely used in the manufacture of metal-cutting tools.

Let us consider the specifics of the elemental composition of the deposited coating (Figure 3). Let us consider the upper layer of the (WC + TiC + Co) substrate at a distance of about 100 nm from the surface (Figure 3, Area 1). In this area, there were such elements in the composition of the carbide, as W and Ti. The presence of a small amount of Cr could be explained by the diffusion of this element from the inner layers of the coating. While considering the composition of the transition layer of the coating (Figure 3, Area 2), it was possible to note the presence of Cr and the small presence of Ti (it could be explained by the diffusion both from the carbide substrate and from the wear-resistant layer). In the layers of the wear-resistant layer, immediately adjacent to the transition layer (Figure 3, Area 3), there was a noticeable presence of Cr and Ti with a small Al content. Upon the further

deposition of the coating, a gradient increase in the Al content occurred, and that could be noted in the elemental analysis data (Figure 3, Areas 4 and 5; with Area 5 located at a distance of about 1 μm from the boundary with the transition layer).

The coating was textured, the intensities of the reflections of the diffraction pattern of the coating were different from the theoretical spectrum. It was also possible that there was the presence of micro- and macrostress, which leads to the broadening and displacement of reflexes. Broadening of the reflections on the diffraction pattern can also be obtained on the nanoscale microstructure of the coating.

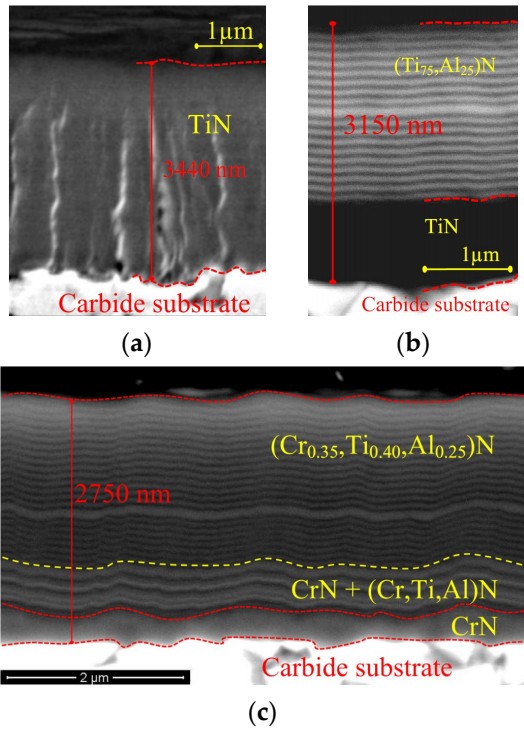

**Figure 1.** Microstructure of the TiN (**a**), TiN-(Ti$_{75}$Al$_{25}$)N (**b**) and Cr-CrN-(Cr$_{0.35}$Ti$_{0.40}$Al$_{0.25}$)N (**c**) coatings (SEM).

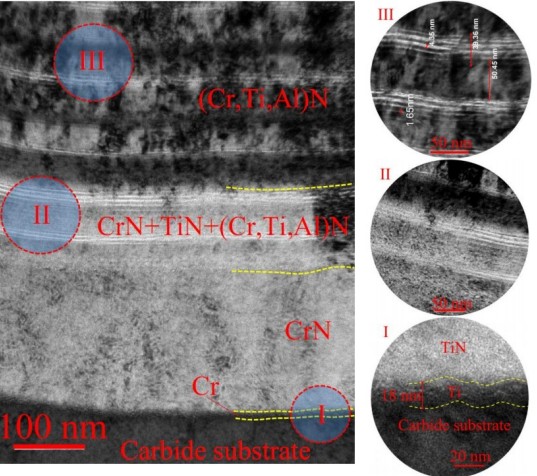

**Figure 2.** Microstructure of the Cr-CrN-(Cr$_{0.35}$,Ti$_{0.40}$,Al$_{0.25}$)N coating (TEM).

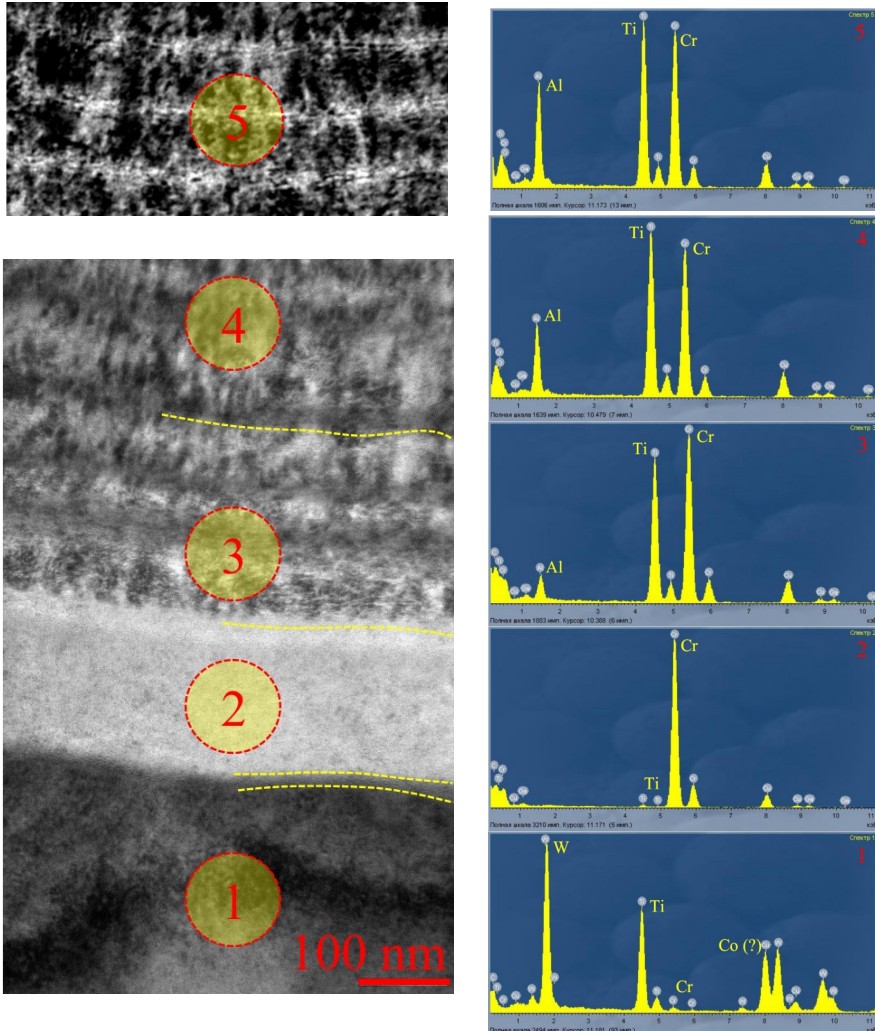

**Figure 3.** Results of the investigation of the elemental composition of the sample with the Cr-CrN-(Cr$_{0.35}$,Ti$_{0.40}$,Al$_{0.25}$)N coating.

The conducted phase analysis of the wear-resistant layer of the coating (Figure 4) showed the presence in the coating of such phases as CrTiN$_2$, AlTiN$_2$, (Cr,Ti,Al)N. The sample included two phases of complex nitrides AlTiN$_2$ and CrTiN$_2$, solid solution (Cr,Ti,Al)N and reflexes of the WC substrate.

Solid solution (Cr,Ti,Al)N had a cubic NaCl-type structure, the space group Fm-3m (Pearson symbol cF8), and the crystal lattice parameter of 4.2 Å.

The complex nitride of CrTiN2 had a cubic structure of NaCl-type, space group Fm-3m (Pearson symbol cF8), and the crystal lattice parameter was 4.184 Å. The complex nitride of AlTiN$_2$ had a cubic structure of NaCl-type, space group Fm-3m (Pearson symbol cF8), and the crystal lattice parameter was 4.194 Å.

The lines of the theoretical spectra of complex nitrides were located very close to each other. In view of the broadening and displacement of the reflexes and the textured coating, it was impossible to resolve the reflexes of these nitrides. It was also possible that the mixture of these two nitrides was present. It was formed on the basis of TiCrN$_2$ nitride and it is a solid aluminum solution based on TiCrN$_2$ nitride, since Al is highly soluble in Ti.

The WC substrate phase had a hexagonal structure, space group P-6m2, and the crystal lattice parameters were 2.907 and 2.837 Å.

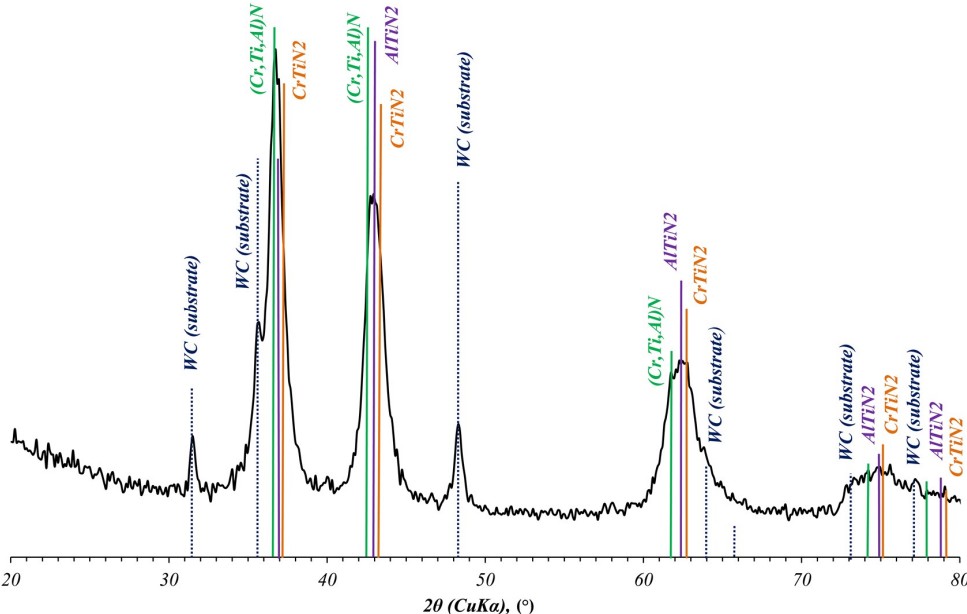

**Figure 4.** X-ray diffraction (XRD) pattern of the Cr-CrN-(Cr,Ti,Al)N coatings on a WC-Co substrate.

Figure 5 presents the TEM analysis of the crystal structure of the coating under study. SAD aperture is shown as a dotted circle in Figure 5. The layer of CrN consisted of large grains, as evidenced by the selected area diffraction patterns (SAED) 1 in Figure 5. The overlying layer (Cr,Ti,Al)N coatings had a polycrystalline structure, SAED pattern 2 Figure 5. There was also the presence of texture maxima. SAED pattern 2 was defined as the space group Fm-3m.

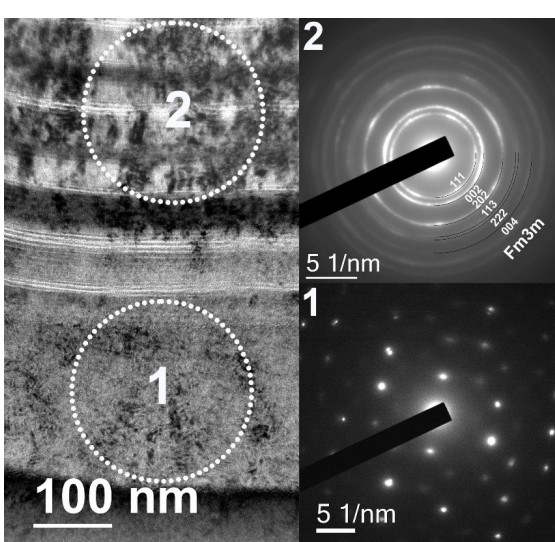

**Figure 5.** Crystalline structure of the sample with the Cr-CrN-(Ti,Cr,Al)N coating.

The microhardness of the coating measured at the load of 20 mN was $32 \pm 1.4$ GPa. The measurement was carried out at 7 points, with the determination of the average value.

Figure 6 shows a panorama of a scratch on the Cr-CrN-(Ti,Cr,Al)N coating. The first signs of the coating failure appeared at the load of $L_{c1}$ = 19 N (zone I on Figure 6a). There were signs of brittle fracture, such as transverse cracks (A in Figure 6b) and areas of coating delamination on the scratch bottom (B1, B2 in Figure 6b). There were areas of plastic deformation of the coating (C1, C2, C3 in Figure 6b), as well as cracks and coating delamination on the edges of the scratch (D in Figure 6b). During the tests at the maximum load (40 N), no complete coating failure occurred. At the end of the

scratch (Zone II in Figure 6a), there were also signs of both brittle and ductile fracture: Transverse cracks (A in Figure 6c), areas of coating delamination of the scratch bottom (B1, B2, B3 in Figure 6c), areas of plastic deformation of the coating (C in Figure 6c), and cracks and coating delamination on the edge of the scratch (D1, D2 in Figure 6c). There was slight chipping of the coating (E in Figure 6c); however, in that area, the coating did not fail to the substrate.

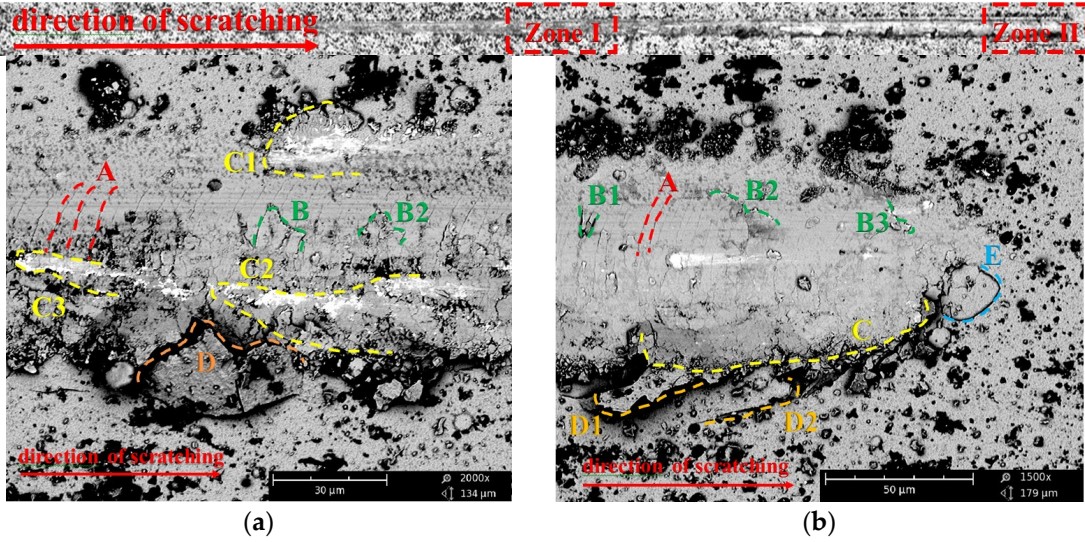

(**a**)　　　　　　　　　　　　　　　　(**b**)

**Figure 6.** Panorama of the scratch on the coating on the sample and scratch fragments at the place where the coating starts to fail (**a**): Zone I and at the end of the scratch (**b**): Zone II; A—transverse scratches in the coating; B1, B2, B3—areas of coating delamination on the scratch bottom; C, C1, C2, C3—areas of plastic deformation of the coating; D, D1, D2—areas of coating delamination on the scratch edges, E—coating chipping.

The results of the cutting tests in dry turning of steel 1045 are presented in Figure 7. If at the cutting speed of $v_c$ = 200 m·min$^{-1}$, tools with the Cr-CrN-(Ti,Cr,Al)N and (Ti,Al)N coatings showed fairly close results, then with an increase in the cutting speed, the difference in tool life periods for the given samples increased, that may have indicated a higher thermostability and oxidation resistance of the Cr-CrN-(Ti,Cr,Al)N coating in combination with the formation on its surface of a chromium oxide-based trioactive film [38].

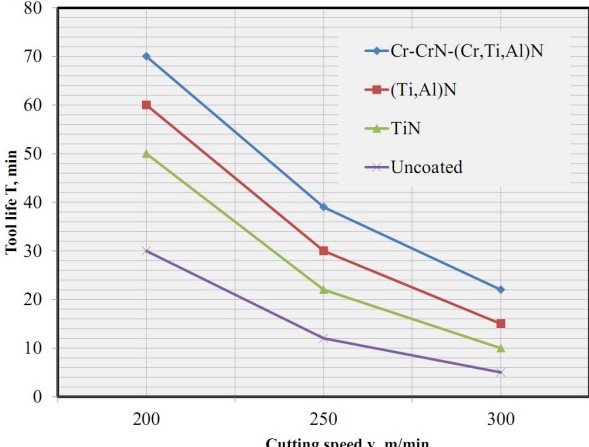

**Figure 7.** Results of cutting tests in dry turning of steel 1045 ($v_c$ = 200, 250 and 300 m·min$^{-1}$; $f$ = 0.2 mm·rev$^{-1}$; $a_p$ = 1.0 mm).

Let us consider the wear kinetics of a tool with the Cr-CrN-(Ti,Cr,Al)N coating under study at different cutting speeds (Figures 8–10). At the cutting speed of $v_c$ = 200 m·min$^{-1}$ (Figure 8), there was a noticeable formation of only a small wear center on the rake face, the wear was balanced, and flank wear is a limiting criterion.

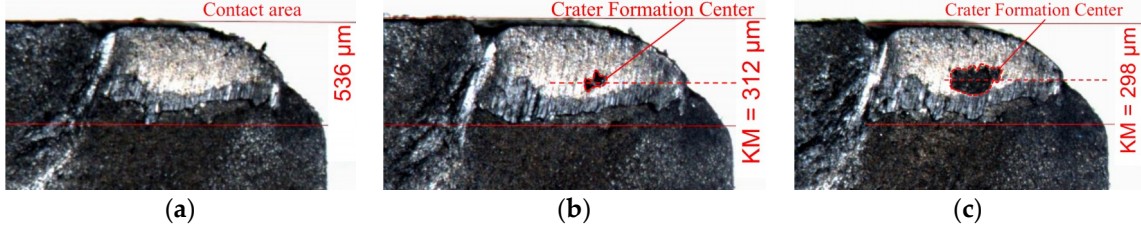

(**a**) (**b**) (**c**)

**Figure 8.** Wear kinetics for a tool with the Cr-CrN-(Ti,Cr,Al)N coating at $v_c$ = 200 m·min$^{-1}$; $f$ = 0.2 mm·rev$^{-1}$; $a_p$ = 1.0 mm; (**a**) 55 min; (**b**) 60 min; (**c**) 70 min of cutting.

At the cutting speed of $v_c$ = 250 m·min$^{-1}$ (Figure 9), the pattern of wear varied. A pronounced crater was formed on the rake face; however, there was a sufficiently wide bridge between the cutting edge and the crater, and the wear stays balanced, with flank wear as a limiting criterion.

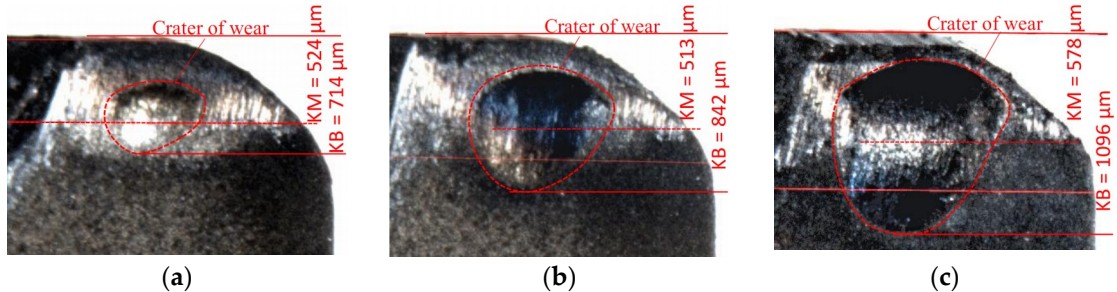

(**a**) (**b**) (**c**)

**Figure 9.** Wear kinetics for a tool with the Cr-CrN-(Ti,Cr,Al)N coating at $v_c$ = 250 m·min$^{-1}$; $f$ = 0.2 mm·rev$^{-1}$; $a_p$ = 1.0 mm; (**a**) 14 min; (**b**) 27 min; (**c**) 39 min of cutting.

With a further increase in the cutting speed up to $v_c$ = 300 m·min$^{-1}$ (Figure 10), the process of crater formation became more active, and a bridge between the cutting edge and the crater failed simultaneously with the achievement of the maximum wear on the flank face. Given that at the specified cutting speed, the tool with the Cr-CrN-(Ti,Cr,Al)N coating showed a sufficiently long tool life, it was clear that the further increase in the cutting speed was not advisable. A formation of an extensive crater may indicate high (close to critical) temperatures in the cutting zone and an active adhesive and oxidative wear. It should be noted that a formation of extensive craters was observed for all the tools under study, while for the uncoated tool and the tool with the TiN coating, rake wear was a limiting criterion at the specified cutting speed.

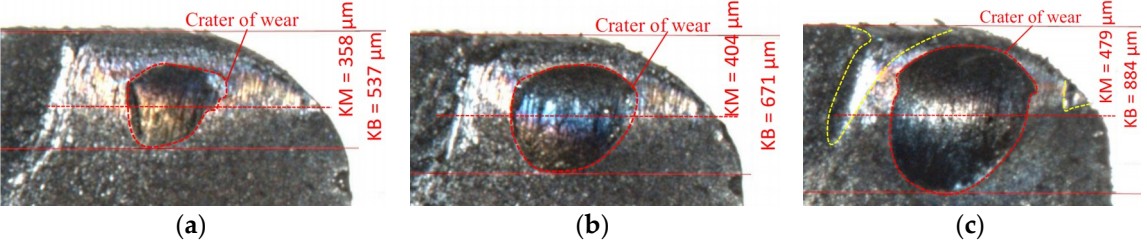

(**a**) (**b**) (**c**)

**Figure 10.** Wear kinetics for a tool with the Cr-CrN-(Ti,Cr,Al)N coating at $v_c$ = 300 m·min$^{-1}$; $f$ = 0.2 mm·rev$^{-1}$; $a_p$ = 1.0 mm; (**a**) 7 min; (**b**) 14 min; (**c**) 22 min of cutting.

Let us consider in more detail the wear process pattern for a tool with the Cr-CrN-(Ti,Cr,Al)N coating at the cutting speeds of $v_c = 200$ (Figure 11) and 250 m·min$^{-1}$ (Figure 12). At the specified cutting speeds, the tool with the coating under study showed a fairly long tool life and a balanced wear pattern. In particular, at the cutting speed of $v_c = 200$ m·min$^{-1}$ (Figure 11), the abrasive wear predominated, there were no signs of brittle fracture of the coating, and the surface of wear was fairly smooth, without chipping and cracking (Figure 11, Area A).

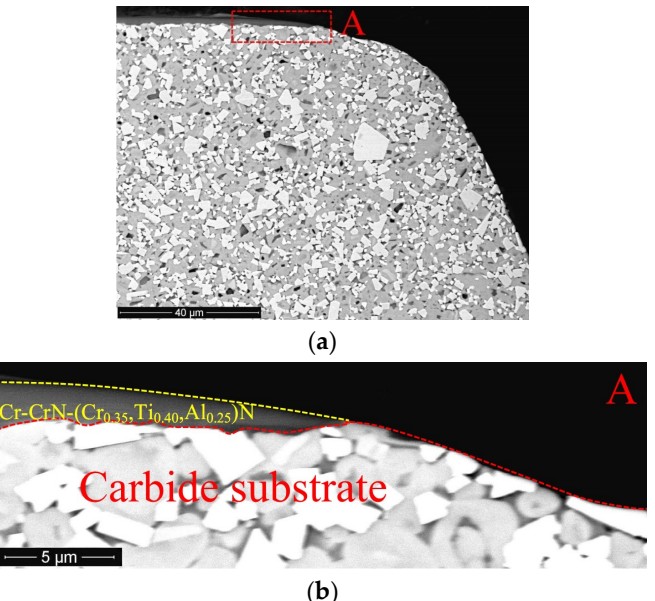

(**a**)

(**b**)

**Figure 11.** Wear pattern on a tool with the Cr-CrN-(Ti,Cr,Al)N coating at $v_c = 200$ m·min$^{-1}$, after 70 min of cutting. General view (**a**); area A (**b**).

At the cutting speed of $v_c = 250$ m·min$^{-1}$ (Figure 12), there were elements of brittle fracture. There was clear chipping of the carbide substrate with penetration of the material being machined into the zone of the coating failure (Figure 12, Area A). In this area, there was also the formation of through transverse cracks in the coating structure, and in some cases, it was possible to see the penetration of elements of the material being machined into a crack (Figure 12, Area D). Meanwhile, the wear process pattern on the crater boundary farthest from the cutting edge (Figure 12, Area B) stayed fairly balanced, without any noticeable signs of brittle fracture and cracking. Let us consider the wear process pattern on the flank face of the tool (Figure 12, Area C). There was a wave-like distortion of the substrate surface as a result of plastic microdeformations. Meanwhile, an adherent of the material being machined penetrated between the coating and the substrate, contributing to the coating failure. Meanwhile, there were no noticeable signs of brittle fracture of the coating in the specified zone.

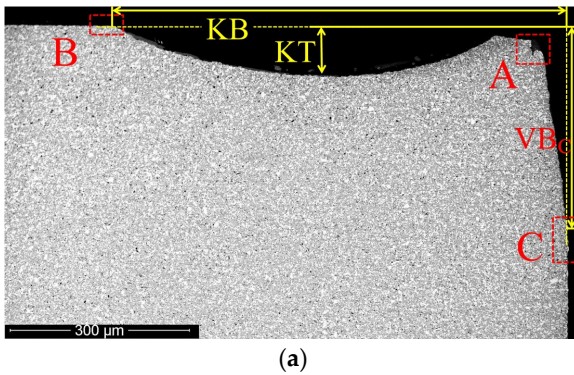

(**a**)

**Figure 12.** *Cont.*

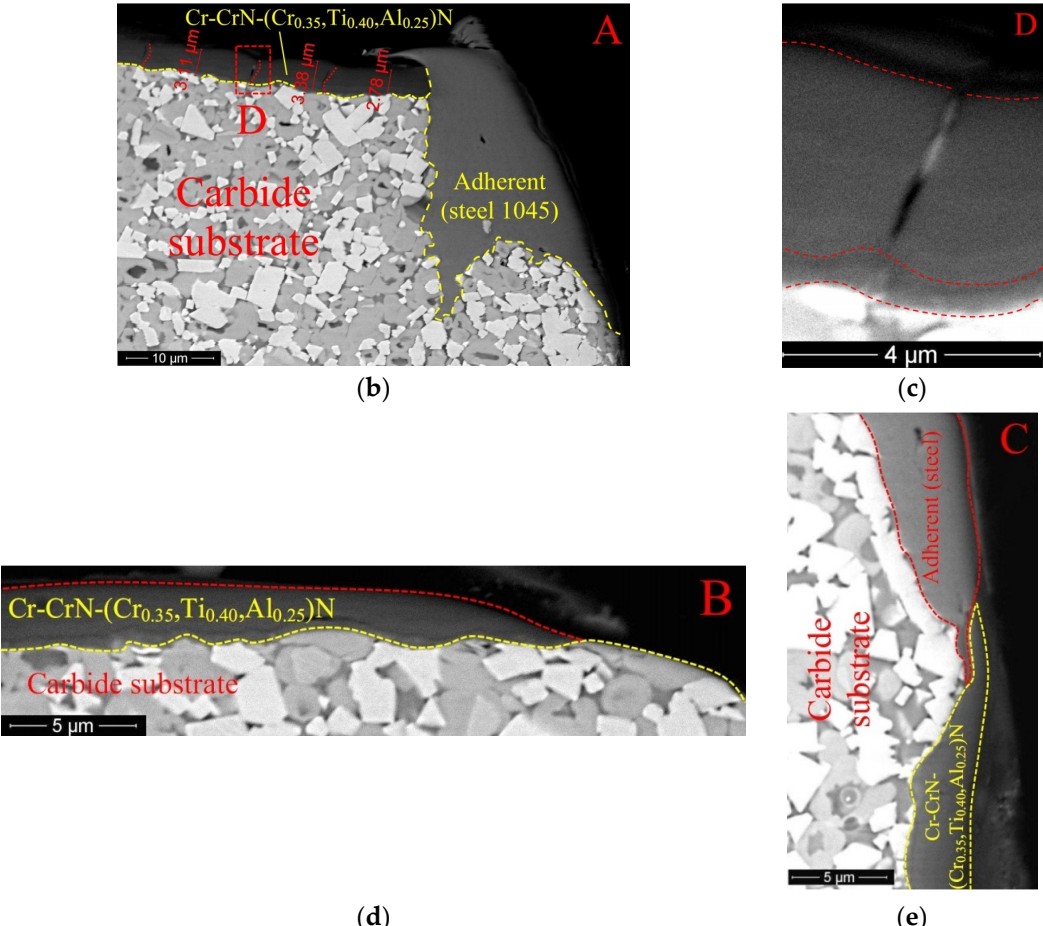

**Figure 12.** Wear pattern on a tool with the Cr-CrN-(Ti,Cr,Al)N coating at $v_c$ = 250 m·min$^{-1}$, after 39 min of cutting. General view (**a**); area A (**b**); area D (**c**); area B (**d**); area C (**e**).

## 4. Conclusions

This study included the investigation into the microstructure, the elemental and phase composition, and the wear process pattern in scratch testing. The microhardness of the Cr-CrN-(Cr$_{0.35}$Ti$_{0.40}$Al$_{0.25}$)N coating was also defined. The cutting tests of the tool with the coating under study were carried out in the dry cutting of steel 1045 at the cutting speeds of $v_c$ = 200, 250, and 300 m·min$^{-1}$.

After the analysis of the obtained results, it was possible to draw the following results:

- The wear-resistant layer of the coating was nano-structured. The thickness of subnanolayers in this layer was 1.5–5.0 nm with the thickness of the binary nanolayer of $\lambda$ = 102 nm.
- The surface layers of the substrate included signs of Cr diffusion from the coating composition.
- The Al content in the composition of the wear-resistant layer showed a gradient increase over about 1 μm from the border with the transition layer.
- The sizes of the crystals in the nano-structured wear-resistant layer were significantly smaller than in the transition layer, and that could be explained by the effect of the nanolayer thickness.
- The conducted phase analysis of the wear-resistant coating showed the presence of such phases as TiN, CrTiN$_2$, and Cr$_2$N.
- The coating showed good resistance of the coating to failure in scratch testing ($L_{c2}$ > 40 N), with signs of both fragile fracture and ductile failure.
- The tool with the coating under study showed the longest tool life in the turning of steel 1045 compared to the uncoated tool and the tools with the commercial TiN and (Ti,Al)N coatings of similar thickness.

- During turning at a cutting speed of $v_c$ = 200 m·min$^{-1}$, a balanced tool wear took place, without any noticeable crater formation and signs of brittle fracture.

- During turning at a cutting speed of $v_c$ = 250 m·min$^{-1}$, the formation of a crater was observed; however, the rake wear was not a limiting criterion. There were signs of brittle fracture and formation of cracks in the coating structure.

- During turning at a cutting speed of $v_c$ = 300 m·min$^{-1}$, an active crater formation took place, and both flank and rake wear processes were limiting criteria.

Therefore, the tool with the Cr-CrN-($Cr_{0.35}Ti_{0.40}Al_{0.25}$)N coating showed the longest tool life at all the cutting speeds under consideration. Meanwhile, a tool with the coating under study can be recommended for use in turning steel 1045 at a cutting speed of $v_c$ = 250 m·min$^{-1}$. At this cutting speed, a tool shows a combination of a rather long tool life and balanced wear process, without any threat of catastrophic wear.

**Author Contributions:** Conceptualization, S.G., A.V. and A.M.; Methodology, S.G. and A.V.; Validation, N.S., F.M., S.S., Y.R. and N.A.; Investigation, S.G. and A.V.; Resources, S.G. and A.V.; Data Curation, A.M.; Writing–Original Draft Preparation, A.V.; Writing–Review & Editing, A.V.; Visualization, S.S.; Supervision, S.G. and A.V.; Project Administration, S.G. and A.V.; Funding Acquisition, A.M. and S.G.

**Funding:** This research was funded by the Ministry of Education and Science of the Russian Federation (No. 9.7886.2017/6.7).

**Conflicts of Interest:** The authors declare no conflict of interest.

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
