# Peer review of "Investigation into Performance of Multilayer Composite Nano-Structured Cr-CrN-(Cr0.35Ti0.40Al0.25)N Coating for Metal Cutting Tools"

_coatings, doi:10.3390/coatings8120447_

Reviewer 1 Report

This paper investigated the wear resistance, thermal stability, tool life of multilayer composite nano-structure Cr-CrN-(Cr,Ti,Al)N coating and showed that it is suitable for metal cutting tools.

Also, author performed many experiments such as XRD, SEM, TEM, cutting test, wear resistance test. I have some comments for publication to this journal as written below.

Major

1) In this paper, authors mentioned many information about tool life and wear resistance. However I think it is suitable for performance than property. So, authors is worth reconsidering the title of paper. In the title, the word ‘properties’ seems to be confused with the data in the manuscript.

2) 2-1. The PVD coating on the key word is little broad, so I think that it would be better to write the FCVAD coating used by the author in the key word.

2-2. It is also reconsidered that the mention or content of thermo stability in the paper is not enough to include in the keywords. Therefore, it is necessary to reconsider about the keywords.

3) Abstract. 3-1. In the abstract, authors mentioned about oxidation resistance, but in the Results section, results and discussions are not enough and clear. Of course, authors explained it as wear behavior. However, wear behavior is an another property. Therefore, authors need to take a more consideration of the consistency of the content.

4) In introduction section, authors reported  some researches. It would be helpful unless authors mentioned the summary of them. All the report is not necessary in the industry. The contents of introduction are long and I feel it is not easy to read due to the writing. I think that it would be better to summarize the contents and references can take their role.

5) In Figure 4. 5-1. Authors mentioned crystal structures and lattice parameters of TiN, CrTiN2, Cr2N. Although it is related to XRD data, it is little ambiguous. Therefore, additional experiments will be necessary such as XPS, EELS, etc.

5-2 Author mentioned that the reason why the Al peak does not appear in the XRD data is because of the α-Ti-based solid solution. It is not enough to explain it. Titanium has the limitation of solubility and aluminum is the one of main elements of the coating layer as EDS showed in the manuscript. The intensity of α-Ti in the pattern is hard to understand as the solid solution. Authors need to require additional explanation about it.

6) 6-1. In Figure 5, there are two SAED patterns. However, the explanation about them is not enough to understand.

6-2. I can’t find the TEM and SAED information in the experimental section. Experimental section is important to understand the manuscript. Authors need to check the experimental section again.

5) In Figure 7, authors compared commercial coatings and the multilayer coatings. However, the reason why authors compared them with the multi-layer is unclear. And also, authors used multi-layer, but commercial coatings seem to be single layer. For comparison, each conditions are not suitable. If authors want to compare it, I think that coating layer should be multi-layer and need the specification of commercial coatings.

Minor

1) In the 124th sentence, after the brittle fracture, a period was omitted.

2) In Table 1, Zr-ZrN-(Zr,Cr,Al)N part must be modified.

3) In figure6, authors explained Figure 6 (a,b,c). However, the Figure 6 (a,b,c) are unclear and need modifications.

4) Author needs a full inspection of the figure once again.

Author Response

Comments and Suggestions for Authors

This paper investigated the wear resistance, thermal stability, tool life of multilayer composite nano-structure Cr-CrN-(Cr,Ti,Al)N coating and showed that it is suitable for metal cutting tools.

Also, author performed many experiments such as XRD, SEM, TEM, cutting test, wear resistance test. I have some comments for publication to this journal as written below.

Major

1) In this paper, authors mentioned many information about tool life and wear resistance. However I think it is suitable for performance than property. So, authors is worth reconsidering the title of paper. In the title, the word ‘properties’ seems to be confused with the data in the manuscript.

The authors agree with the recommendation of the reviewer. The question of translation. Replaced by “performance”.

2) 2-1. The PVD coating on the key word is little broad, so I think that it would be better to write the FCVAD coating used by the author in the key word.

The authors agree with the recommendation of the reviewer. In principle, FCVAD is a variation of the PVD method, but a more specific name is FCVAD.

2-2. It is also reconsidered that the mention or content of thermo stability in the paper is not enough to include in the keywords. Therefore, it is necessary to reconsider about the keywords.

The authors agree with the recommendation of the reviewer, “thermo stability” is replaced by “tool life”.

3) Abstract. 3-1. In the abstract, authors mentioned about oxidation resistance, but in the Results section, results and discussions are not enough and clear. Of course, authors explained it as wear behavior. However, wear behavior is an another property. Therefore, authors need to take a more consideration of the consistency of the content.

The authors carefully studied the text of the abstract, they think that in the abstract there is no mention of “oxidation resistance”, since this issue has not been studied in detail in this article.

4) In introduction section, authors reported  some researches. It would be helpful unless authors mentioned the summary of them. All the report is not necessary in the industry. The contents of introduction are long and I feel it is not easy to read due to the writing. I think that it would be better to summarize the contents and references can take their role.

Introduction significantly reduced.

 5) In Figure 4. 5-1. Authors mentioned crystal structures and lattice parameters of TiN, CrTiN2, Cr2N. Although it is related to XRD data, it is little ambiguous. Therefore, additional experiments will be necessary such as XPS, EELS, etc.

The authors agree with the reviewer’s comment. Additional studies were conducted and the results of past studies were reanalyzed. In particular, phase A can be classified as “possible” phases, since only one strong peak is superimposed and there are no other peaks, based on this it is incorrect to conclude that it is present. Accordingly, this phase has been removed from the XRD diagram. Since the corrected XRD data (Figure 4) now correlates well enough with the SAED data (Figure 5), the authors believe that they can be considered reliable. Corrected versions of Figures 4 and 5, as well as corrected descriptions added to the article.

The data on the crystal structure of the identified phases are taken from the corresponding cards in the ICCD PDF-2 database. As an explanation, information about parameters of crystal lattices and the crystal structure was given.

5-2 Author mentioned that the reason why the Al peak does not appear in the XRD data is because of the α-Ti-based solid solution. It is not enough to explain it. Titanium has the limitation of solubility and aluminum is the one of main elements of the coating layer as EDS showed in the manuscript. The intensity of α-Ti in the pattern is hard to understand as the solid solution. Authors need to require additional explanation about it.

 Authors assume the presence of two complex nitrides in the coating: TiCrN2 and AlTiN2, as well as their mixture in the transition zones of one sublayer to another. Presumably, a mixture of these nitrides is formed on the basis of TiCrN2 nitride and it is a solid aluminum solution based on TiCrN2 nitride, because Al well dissolved in Ti.

Previously, the presence of AlTiN2 nitride was not detected, because both complex nitrides have a cubic syngony (space group Fm-3m) and close lattice parameters, about 4.18 - 4.19 Å.

The lines of the theoretical spectra of these nitrides are located very close to each other. The theoretical spectra of the nitrides TiCrN2 and AlTiN2 superimposed on the diffraction pattern of the coating are presented below. As you can see, the lines almost overlap.

The coating is textured, therefore the intensities of the reflections of the coating diffractogram are different from the theoretical spectrum. It is also possible the presence of micro-and macrostress, which leads to the broadening and displacement of reflexes. In addition, the coating is nanostructured, which leads to broadening of the reflections on the diffractogram. Therefore, it is not possible to resolve the reflexes of these two nitrides. But the presence of aluminum is revealed exactly.

Also in the coating there is a phase of a solid solution (Cr, Ti, Al) N, in which the stoichiometric composition includes aluminum.

6) 6-1. In Figure 5, there are two SAED patterns. However, the explanation about them is not enough to understand.

Data decryption added.

6-2. I can’t find the TEM and SAED information in the experimental section. Experimental section is important to understand the manuscript. Authors need to check the experimental section again.

Information supplemented.

 5) In Figure 7, authors compared commercial coatings and the multilayer coatings. However, the reason why authors compared them with the multi-layer is unclear. And also, authors used multi-layer, but commercial coatings seem to be single layer. For comparison, each conditions are not suitable. If authors want to compare it, I think that coating layer should be multi-layer and need the specification of commercial coatings.

Description of commercial coatings added. Their microstructural study was done using SEM. The first coating is monolayer TiN, the second is a two-layer coating with a nanostructured outer layer. The total coating thickness is the same. The authors believe that a more detailed study of these commercial coatings does not make sense, since they are not the main objects of research.

Minor

1) In the 124th sentence, after the brittle fracture, a period was omitted.

 Typo fixed.

2) In Table 1, Zr-ZrN-(Zr,Cr,Al)N part must be modified.

Typo fixed.

 3) In Figure 6, authors explained Figure 6 (a,b,c). However, the Figure 6 (a,b,c) are unclear and need modifications.

Changes have been made to the Figure for the best understanding of its content.

4) Author needs a full inspection of the figure once again.

The figures are checked and in some cases changed.

The authors are grateful to the Reviewer for serious attention to the manuscript. A number of valuable recommendations of the reviewer were taken into account by the authors when working with the revised version. These recommendations will also help the authors in their future work. Thanks again!

Author Response

In my opinion, the manuscript should be substantially revised in terms of consistency, better explanations of the essentials, and using ISO terminology.

A clear objective of the manuscript should be stated at the end of Introduction instead of summary of the research experience available. Moreover, the introduction should not only list what is available today and how good it is but rather it should point out some problems to be solved so that the manuscript objective should be logically formulated as an attempt to solve the revealed/discussed problems.

The essence of the design of a special coating is to much the conditions at the tool-chip and tool workpiece interfaces. To do that, one needs to know and understand these conditions in terms of contact stresses, temperatures, and relative velocity over the listed interfaces. Unfortunately, the authors did not make any attempt to discuss these conditions, i.e. what a designed coating is for. The relevant information can be found in

Astakhov, V.P., Tribology of Metal Cutting, Elsevier: London,  2006.

Astakhov V.P., Tribology of Cutting Tools, Chapter 1 in book: Tribology in Manufacturing Technology, Springer, 2013, pp. 1-66 (available at http://publications.axfree.com/TribologyCTPublished.pdf)

The authors substantially revised and supplemented the Introduction in accordance with the comments and recommendations of the reviewer.

It is stated “It can be predicted that such a coating will provide a balanced combination of wear resistance, thermo stability, resistance to high temperature oxidation, and brittle fracture.” – You can’t predict it – you just can anticipate it based on previous results/scientific rationale etc.

Of course, it is the “anticipated” not “predicted” that is appropriate here. Inaccuracy of translation. Fixed.

Test conditions are poorly indicated:

Hardness of the work material (steel 1045) should be indicated.

Information added.

Cutting tool geometry (insert type, rake, clearance, inclination and tool cutting edge angles of the major and minor cutting edges, insert nose radius, the radius of the cutting edge) should be given.

Information added.

Tool material (at least grade and make) should be given.

Information added.

The dimensions of the workpiece (length and initial diameter) should be given.

Information added.

Coating thickness measurement should be included.

This information is contained in the results of microstructural studies (Figure 1).

In evaluation of the crater wear, parameters of the crater KT, KM, and KB should be reported according to standard ISO 3685:1993

In accordance with the recommendations of the reviewer, additions have been made to Figures 8-10 and 12. The KT value was not measured during the cutting process, therefore its values are absent. This value is indicated on the transverse section (Figure 12).

The scale of Fig 7 should be changed as at its present appearance it is not clear the difference in tool life between the listed coatings. It may be within a scatter of test results.

The scale is changed from logarithmic to ordinary numeric.

The authors are grateful to the Reviewer for a number of important recommendations that significantly improved the quality and significance of this manuscript. Undoubtedly, a deeper understanding of the cutting process and contact processes in the cutting zone helps to better understand the algorithm for choosing a coating of rational composition and architecture.

Round  2

Reviewer 1 Report

English spell check required

Reviewer 2 Report

The manuscript can be accepted as the authors addressed my concerns.